# Discovery of Novel miRNAs in Colorectal Cancer: Potential Biological Roles and Clinical Utility

**DOI:** 10.3390/ncrna9060065

**Published:** 2023-10-26

**Authors:** Iael Weissberg Minutentag, Ana Laura Seneda, Mateus C. Barros-Filhos, Márcio de Carvalho, Vanessa G. P. Souza, Claudia N. Hasimoto, Marcelo P. T. Moraes, Fabio A. Marchi, Wan L. Lam, Patricia P. Reis, Sandra A. Drigo

**Affiliations:** 1Department of Surgery and Orthopedics, Medical School, São Paulo State University (UNESP), Botucatu 18618-687, Brazil; ana.seneda@unesp.br (A.L.S.); claudia.hasimoto@unesp.br (C.N.H.);; 2Experimental Research Unity (UNIPEX), São Paulo State University (UNESP), Botucatu 18618-687, Brazil; vg.souza@unesp.br; 3Centro Internacional de Pesquisa (CIPE)—A. C. Camargo Cancer Center, São Paulo 01508-010, Brazil; 4School of Veterinary Medicine and Animal Science, São Paulo State University (UNESP), Botucatu 18618-687, Brazil; 5Department of Genetics, Institute of Biosciences, São Paulo State University (UNESP), Botucatu 18618-687, Brazil; 6Department of Pathology, Medical School, São Paulo State University (UNESP), Botucatu 18618-687, Brazil; 7Department of Head and Neck Surgery, Medical School and São Paulo State Cancer Institute (ICESP), University of São Paulo (USP), São Paulo 01246-903, Brazil; 8British Columbia Cancer Research Centre, Vancouver, BC V5Z 1L3, Canada

**Keywords:** colorectal cancer, tumor location, novel microRNA, miRMaster, tissue specificity, prognosis, small-RNAseq

## Abstract

Deregulated miRNAs are associated with colorectal cancer (CRC), with alterations depending on the tumor location. Novel tissue-specific miRNAs have been identified in different tumors and are associated with cancer. We used miRMaster to identify novel miRNAs in CRC from the TCGA and GEO data (discovery and validation groups). We used TCGA data from five tissues to analyze miRNA tissue specificity. miRDB was used to predict miRNA targets, and the UCSC Xena Browser was used to evaluate target expression. After successive analyses, we identified 15 novel miRNAs with the same expression patterns in CRC in both the discovery and validation groups. Four molecules (nov-miR-13844-5p, nov-miR-7154-5p, nov-miR-5035-3p, and nov-miR-590-5p) were differentially expressed in proximal and distal CRC. The nov-miR-3345-5p and nov-miR-13172-3p, which are upregulated in tumors, are only expressed in colorectal tissues. These molecules have been linked to a worse prognosis in right-sided colon and rectal carcinomas. An analysis revealed an association between eight novel miRNAs and 81 targets, mostly cancer-related genes, with varying expression based on tumor location. These findings provide new miRNAs with potential biological relevance, molecular biomarkers, and therapeutic targets for CRC treatment.

## 1. Introduction

Colorectal cancer (CRC) is the third most common cancer and the second leading cause of cancer-related deaths [1]. In Brazil, CRC is the second most common cancer in both men and women, with an estimated 45,630 new cases nationwide by the year 2023 [2]. CRC is a heterogeneous disease with a large amount of evidence showing that proximal and distal CRCs are distinct entities with different embryological origins, characteristic immune profiles, genetic alterations, and different clinical outcomes [3]

Non-coding RNAs (ncRNAs) have historically been considered merely intermediaries between DNA and proteins, except for ribosomal and transfer RNA. However, our transcriptome is composed of only 2% coding RNA, whereas the remaining 98% consists of non-coding RNAs that play diverse roles in post-transcriptional regulation [4]. MicroRNAs (miRNAs or miRs) are small ncRNA molecules with approximately 22 nucleotidesin length associated with epigenetic silencing, participating in various cellular processes, such as growth, differentiation, development, and apoptosis [5]. Numerous studies have provided evidence of miRNA involvement in colorectal carcinogenesis, including tumor initiation, progression, and metastatic spreading, revealing potential diagnostic and prognostic biomarkers for CRC, as well as therapeutic targets [6].

Currently, nearly 2800 human miRNAs are annotated in public repositories; however, more recent studies have shown a higher number of miRNAs expressed in the human genome than previously estimated. Human miRNA repertoire characterization was mostly limited to low-coverage techniques used in early studies, the detection of miRNA sequences that are conserved across species and different tissue types, and the predominant detection of more abundant transcripts [7,8]. However, recent analyses using next-generation sequencing (NGS) have revealed new human miRNAs with higher tissue specificity and lower expression levels than known annotated miRNAs [8]. By using advanced bioinformatics tools for NGS of small RNAs (small-RNAseq), 2469 new candidates for human miRNAs have been reported, of which 1098 have been validated [7]. Other studies have reported novel miRNAs associated with different tumor types, such as head and neck cancer [9], gastric adenocarcinoma [10], papillary thyroid carcinoma [11], and cervical cancer [12]. Taken together, these data indicate a greater diversity of human miRNAs than previously anticipated, revealing tissue-specific regulatory networks associated with these novel molecules relevant to tumor initiation and progression and extending the miRNA repertoire for molecular biomarkers and drug targets in cancer.

Owing to their high tissue specificity and role in tumorigenesis, miRNAs have also been indicated as molecular biomarkers for diagnosing cancer and predicting patient outcomes. Dysregulation of miRNAs has been reported in CRC and plays an important role in its development and metastasis. Despite recent high-throughput and highly sensitive sequencing methods that have allowed for the discovery of novel miRNAs in different tumors with potential clinical utility, there is a paucity of studies addressing unannotated tissue-specific miRNAs in CRC. Thus, in this study, we used large-scale analysis of high-throughput sequencing data to discover novel miRNAs with a role in colorectal carcinogenesis and evaluated their expression patterns according to tumor location, prognosis, and putative target genes.

## 2. Results

### 2.1. Identification of Novel microRNAs

By using a custom discovery pipeline followed by miRMaster analysis, we identified 358 novel microRNA candidates (without overlap with miRBase-annotated miRNAs) in a discovery group composed of a large cohort of colorectal samples (N = 522) from The Cancer Genome Atlas (TCGA (Figure 1, Appendix A). By comparing CRC (n = 511) and normal (n = 11) tissues, 249 out of 358 unannotated miRNAs were detected as differentially expressed (DE) between tumor and normal samples (*p* < 0.05) (Figure 1).

After filtering, we evaluated the GC content, sequence length, folding structures, and genomic distribution of the miRNA candidates relative to annotated miRNAs to verify the likelihood of our candidates being real novel miRNAs. All miRNA candidates showed similar features to annotated miRNAs (Appendix A).

#### In Silico Validation of the Novel miRNAs

To validate the expression of miRNA candidates in CRC, an independent set of colorectal samples (n = 55; GSE89974) was analyzed following the same pipeline and parameters used for the discovery group. A total of 126 putative novel miRNAs were identified. Although the validation group consisted of a limited number of CRC samples (n = 35), we were able to detect 55 DE miRNAs in CRC tissues compared to normal tissues (n = 20) (Figure 1). By comparing the miRNAs identified in TCGA and GSE89974 samples, 21 novel miRNAs were identified in both groups (Appendix A). Fifteen out of twenty-one candidate miRNAs showed similar expression patterns between tumor and normal tissues in both discovery and validation groups, with seven novel miRNAs showing increased expression and eight decreased expression in tumors (Figure 1 and Table 1), suggesting that these candidates are likely to be true miRNAs with a role in colorectal carcinogenesis. Therefore, only 15 validated miRNAs were used in the subsequent analysis (Figure 1).

### 2.2. Genomic Location and Tissue-Specific Expression Patterns of Novel miRNAs

#### 2.2.1. Genomic Location

We evaluated the distribution of known miRNAs, candidate miRNAs, and predicted targets in the genome by using a circle plot. Interestingly, some candidates were mapped to regions rich in known miRNAs: nov-miR-13996-5p (chrX+: 7187376–7187444), nov-miR-7154-5p (chr20-: 38425272–38425332), nov-miR-5035-5p (chr19+: 12952630–12952689), nov-miR-1156-5p/3p (chr11-: 1477615–1477671), nov-miR-13172-3p (chr9+: 115489101–115489152), and nov-miR-8861-5p (chr2-: 69520113–69520174) (Appendix A). These data suggest that these miRNA candidates might be members of known miRNA clusters.

We also assessed whether our 15 novel miRs were localized in chromosomal fragile sites; three candidates are transcribed from these sites: nov-miR-8861-5p on fragile site FRA2E (chr2p13: 68600001–75000000), nov-miR-766-3p on fragile site FRA11C (chr11p15.1: 16200001–21700000), and nov-miR-3345-5p on fragile site FRA15A (chr15q22.2: 59100001–67500000). All three fragile sites were classified as having a common frequency and can be induced by the antibiotic aphidicolin.

#### 2.2.2. Tissue-Specific Expression Patterns of Novel miRNAs

We investigated whether novel miRNA candidates can be used to differentiate tumors from normal samples. Principal component analysis (PCA) was performed using known and novel miRNAs identified in the TCGA group (Figure 2a,b, respectively). Different expression patterns of both known and novel miRNAs were observed between tumor and normal samples, showing separate sample groups.

To determine whether the novel miRNAs exhibited organ-specific expression patterns, we examined 15 validated novel miRNAs in normal colorectal samples from TCGA and GSE89974 groups (n = 31) and normal samples from five different tissues obtained from TCGA (Figure 2c). Different expression patterns of the miRNA candidates were observed in the samples. Interestingly, two out of fifteen novel miRNAs were exclusively expressed in colorectal tissues: nov-miR-13172-3p and nov-miR-3345-5p. Conversely, three novel miRNAs were expressed in all tissues: nov-miR-1156-5p, nov-miR-8861-5p, and nov-miR-13996-5p (Figure 2c). In addition, the expression patterns of the 15 novel miRs clearly distinguished colorectal samples from normal samples derived from other tissues (liver, breast, urinary bladder, brain, and lung), indicating tissue-specific expression patterns of these molecules (Figure 2d).

Next, we investigated whether the novel miRNAs were DE according to the anatomical tumor location. We compared the expression of 15 novel miRNAs in samples from right-sided colorectal cancer (R-CRC), left-sided colorectal cancer (L-CRC), and rectal cancer (RC). Four novel miRNAs were DE according to tumor location. Two of them (nov-miR-13844-5p and nov-miR-7154-5p) were also significantly more expressed in tumors than in normal tissues (Figure 3), suggesting an oncogenic role for these novel miRNAs. The candidate miR nov-miR-13844-5p showed higher expression of L-CRC (*p* = 4.245 × 10^−5^) and RC (*p* = 0.002683) than R-CRC samples (Figure 3a). Higher levels of nov-miR-590-5p were detected in L-CRC tissues than in R-CRC tissues, although it was not differentially expressed in tumors compared to normal tissues (*p* = 0.0257, Figure 3b). Increased expression of nov-miR-7154-5p (*p* = 0.0028) was detected in L-CRC (*p* = 4.245 × 10^−5^) and RC (*p* = 0.002683) samples compared to that in L-CRC samples (Figure 3c). The nov-miR-5035-3p downregulated in tumors compared to normal samples showed lower levels in L-CRC tissues than in R-CRC tissues (*p* = 0.0294, Figure 3d).

### 2.3. Prognostic Relevance

The prognostic value of the 15 novel miRNAs expressed in colorectal tumors was assessed in the TCGA dataset. Two novel miRNAs were significantly associated with overall survival in CRC patients according to the anatomical site (Figure 4). Higher levels of nov-miR-3345-5p in R-CRC were associated with decreased survival (*p* = 0.043) (Figure 4a). In addition, overexpression of nov-miR-7154 in RC was associated with a worse prognosis (*p* = 0.008) (Figure 4b).

### 2.4. Target Prediction and Expression in CRC

Target prediction using the miRDB platform revealed 2412 target genes regulated by 15 novel miRNAs.

To refine our predicted target gene list, we evaluated the expression levels of target genes in the discovery cohort available on the Xena Browser platform. Target genes were selected if they were DE between tumor and normal samples and if they presented opposite expression patterns in relation to their novel miRNA regulators. This analysis revealed 176 target genes regulated by 14 novel miRNAs; 24 genes were upregulated, and 152 were downregulated (Figure 5a,b).

Next, we analyzed the DE target genes according to their anatomical location in R-CRC, L-CRC, and RC (Figure 5a,b). R-CRC samples showed 141 DE-predicted targets, with 17 targets exclusively expressed in these tumors. In L-CRC, 150 DE targets were detected, 13 of which were exclusively detected in L-CRC. In the RC samples, 67 were DE, with 9 exclusively detected in these tumors (Figure 5a,b).

To improve the target prediction, an integrative analysis was performed to select novel miRNA-target gene pairs with a statistically significant negative correlation (r ≤ −0.1 e *p* ≤ 0.05). A total of 81 target genes exhibited a significant correlation with 9 out of 15 candidate miRNAs, with 6 upregulated miRNAs (nov-miR-13172-3p, nov-miR-13844-5p, nov-miR-3345-5p, nov-miR-7154-5p, nov-miR-766-3p, and nov-miR-8861-5p) and 3 downregulated miRNAs (nov-miR-1156-5p, nov-miR-590-5p, and nov-miR-5065-5p) (Appendix A).

It is noteworthy that nov-miR-3345-5p and nov-miR-13172-3p, both upregulated in tumors and expressed exclusively in colorectal tissues, showed 21 putative targets (Appendix A). The upregulated nov-miR-3345-5p, which is also associated with poor prognosis in R-CRC, showed a negative correlation with 10 targets. Among them, *BNC2*, *SYNM*, and *TCF21* target genes were detected in all tumor locations, and three targets (*GRIK2, KLHL14*, and *MS4A2*) were shared by R-CRC and L-CRC but not by RC. Candidate nov-miR-13172-3p showed 11 target genes. The correlation of *AFF3* and *DAAM2* with the candidate miRNAs was detected at all tumor locations.

This analysis also revealed 10 miRNA-target pairs exclusively present in R-CRC, 20 exclusively detected in L-CRC, and 7 pairs in RC (Appendix A). Interestingly, nov-miR-8861-5p, which is upregulated in tumors, presented the highest number of targets in all three anatomic locations, showing both shared and exclusive targets for each location.

The candidate nov-miR-7154-5p was upregulated in CRC, particularly in distal tumors (L-CRC and RC), was associated with poor survival in RC, and showed six predicted targets, with the *EPHA7* gene exclusively detected in RC. Three other candidates (nov-miR-13844-5p, nov-miR-5035-3p, and nov-miR-590-5p) with differential expression between proximal and distal CRC also showed significant correlations with their predicted targets.

The downregulated candidates nov-miR-1156-5p, nov-miR-590-5p, and nov-miR-5065-5p showed different targets according to the tumor location. The nov-miR-1156-5p showed a correlation with *RNF183* exclusively in RC, and nov-miR-590-5p showed a correlation with *HOMER1* and *TBX15* in R-CRC and L-CRC, respectively. The nov-miR-5065-5p was predicted to regulate *ERP27* only in L-CRC samples (Appendix A).

### 2.5. Interaction Networks

Interaction analysis, including the 81 target genes, showed that 47 genes presented functional interactions between them (Figure 6). These genes are regulated by seven candidate miRNAs. *CACNA1B* and *DLG2* showed the highest number of interactions with other genes and were regulated by nov-miR-8861-5p and nov-miR-13172-3p, respectively. The candidate nov-miR-8861-5p emerged as the miRNA that regulates the highest number of genes (32 target genes) with interactions among them. Interestingly, this miRNA regulates unique genes from all three anatomical locations, including CHGB in the right colon, *CACNA1B* and *GLRA3* in the left colon, and *TFAP2B* in rectal cancer. Furthermore, this candidate miRNA was predicted to regulate four interconnected genes: *PEG3*, *MKRN3*, *NPTX1*, and *BEND4* (Figure 6). These findings suggested that nov-miR-8861 plays an important role in colorectal carcinogenesis by regulating different genes and pathways.

## 3. Discussion

In this study, using small-RNA-seq data and analysis using the miRMaster algorithm, we identified 358 new miRNAs in a large cohort of colorectal samples. After curating analysis and validation with an independent dataset, we successfully detected 15 novel miRNAs with potential biological relevance in CRC, as demonstrated by tissue specificity and differential expression according to tumor location and association with prognosis, suggesting that these molecules might be involved in tumor development and progression.

Of the 358 novel miRNAs detected in the discovery group, 249 (70%) were differentially expressed between tumors and adjacent normal tissues, with 118 overexpressed and 131 underexpressed in CRC. The strategy used for the discovery of new miRNAs, followed by validation with an independent dataset, identified a limited but robust result, in which 15 out of 358 candidates (4.2%) were detected as novel miRNAs in CRC. Barros-Filho et al. [11] analyzed papillary thyroid carcinomas and reported 234 novel candidate miRNAs, of which 16 (6.84%) were validated in an independent public dataset. Another study evaluating gastric tumors reported 170 candidates, of which 43 (25.3%) were validated in silico [10]. Validation of 102 novel miRNAs from 280 candidates (36.4%) has been reported in renal carcinoma [13]. Rock et al. [9] validated 102 (69.9%) of the 146 candidates in head and neck carcinomas. This variation can be explained, in part, by the different analysis strategies used in these studies and the larger number of samples in the validation groups, which allowed the detection of a greater number of candidates. In our study, despite the small sample size in the validation group (35 CRC and 20 normal samples) compared with the discovery group (n = 522), 21 miRNAs were identified, with 15 (71.43%) showing consistent expression patterns when comparing tumors to normal tissue samples.

To support these novel miRNA transcripts as true positive miRNA sequences, molecular characterization of the novel miRs was performed, which showed similarities with known miRs, such as GC content, size, and genomic location in regions enriched with known miRs and chromosomal fragile sites. We showed that both the known miRNAs and candidates exhibited GC content above 50%, which is in agreement with a previous report showing that known dysregulated miRNAs in CRC exhibit an average GC content above 50%, indicating that cancer-regulated miRNAs might be of the non-conserved type and enriched in cytosine and guanine [14].

The validated miRNAs were mapped to chromosomes 2, 3, 9, 10, 11, 13, 15, 19, 20, and X, most of which were located in regions rich in known miRNAs. Three novel miRNAs (nov-miR-3345-5p, nov-miR-766-3p, and nov-miR-8861-5p) were detected in known genomic fragile sites, FRA2E, FRA11C, and FRA15A, all of which are classified as common fragile sites (CFS) because they are present in most of the human population [15]. Fragile sites are gaps, constrictions, or breaks in metaphase chromosomes that arise when cells are exposed to perturbation of the DNA replication process and are linked to regions of chromosomal rearrangement in cancer [16]. Fragile sites are also particularly dense in miRNAs, many of which are located within genes commonly translocated in cancers [17]. Overall, these data provide additional evidence that these candidates are true-positive miRNAs.

In addition, we observed that similar to observations with known miRNAs, the 15 novel miRNAs detected here could differentiate colorectal tumors from adjacent normal tissues, thus enhancing the repertoire of miRNAs in CRC. Several miRNAs exhibit specific expression patterns in tumors such as lung, breast, brain, liver, and colorectal cancers, making them potentially valuable for the development of new therapies [18]. In addition to this ability to differentiate tumors from normal tissues, it is highly desirable that potential therapeutic targets present tissue specificity. Two out of the fifteen candidates detected here, nov-miR-13172-3p and nov-miR-3345-5p, were exclusively expressed in colorectal tissues and are potential candidates for further experimental validation.

Among the 15 novel miRNAs detected in colorectal samples, 9 candidates (nov-miR-1156-3p, nov-miR-13172-3p, nov-miR-13844-5p, nov-miR-3345-5p, nov-miR-5035-3p, nov-miR-5065-5p, nov-miR-7154-5p, nov-miR-766-3p, and nov-miR-8999-3p) showed significant differential expression between tumor and normal tissues. The nov-mir-13844-5p was not detected in normal samples from the discovery group, although it was detected in normal samples from the validation group. This suggests that this candidate miRNA may be highly specific to the tumor context, potentially making it a promising target for diagnostic purposes [19].

Several miRNAs have been reported to have distinct expression profiles according to tumor location [20,21,22,23,24]. In our study, four candidates (nov-miR-13844-5p, nov-miR-590-5p, nov-miR-7154-5p, and nov-miR-5035-3p) were DE between anatomical sites. Interestingly, similar expression patterns of these candidates were observed in L-CRC and RC compared to R-CRC, corroborating previous reports showing that rectal tumors have a molecular profile that is more closely aligned with L-CRC than with tumors from other intestinal sites [25].

To assess whether the 15 novel miRNAs could be associated with CRC prognosis, survival curves were constructed based on their transcript levels in tumors compared to normal samples. Two candidates, nov-miR-3345-5p and nov-miR-7154-5p, both upregulated in tumor samples, were associated with poor prognosis in R-CRC and RC, respectively, suggesting that these novel miRNAs could be considered promising prognostic biomarkers in CRC of different tumor locations. Novel miRNAs were also associated with prognosis in thyroid cancer [11], gastric cancer [10], and head and neck tumors [9], reinforcing that the unannotated miRNAs, such as those described here, may play a role in tumor progression and can improve the prognostic risk stratification of colorectal tumors.

To better understand the biological role of the novel miRNAs discovered in CRC, target prediction followed by integrative analysis evaluating novel miRNA-target gene pairs was performed. Numerous genes strongly implicated in colorectal carcinogenesis were predicted to be targets of our candidate miRNAs, which will be explored further in depth. Candidates nov-miR-3345-5p and nov-miR-13172-3p, detected here as colorectal-specific miRNAs, showed 21 putative targets. *TCF21* gene showed a negative correlation with nov-miR-3345-5p in tumors from all anatomical locations. *TCF21,* a transcription factor that regulates cellular differentiation and organogenesis, has been detected as a tumor suppressor in CRC, and reduced *TCF21* expression is associated with poor prognosis. This gene mediates inactivation of the PI3K/AKT pathway and inhibits the action of metalloproteinases in CRC [26]. Additionally, nov-miR-3345-5p was predicted to regulate *SYNM*, which encodes for synemin, a type IV intermediate filament. This gene has been suggested to be a tumor suppressor in breast cancer and is regulated by methylation [27]. *AFF3* was identified as a putative target of nov-miR-13172-3p in CRC. *AFF3* encodes a nuclear transcriptional activator. AFF3, among other RNA-binding proteins, has been investigated in CRC. Downregulation of *AFF3* has been detected in CRC and is associated with poor prognosis [28]. Further experimental validation is required to prove the regulation of these genes by nov-miR-3345-5p and nov-miR-13172-3p in colorectal tumors.

Four candidates (nov-miR-13844-5p, nov-miR-7154-5p, nov-miR-5035-3p, and nov-miR-590-5p) showed different expression patterns according to tumor location. The candidate nov-miR-13844-5p, identified as upregulated in distal tumors compared with proximal tumors, revealed different targets according to these locations. Possible targets identified for nov-miR-13844-5p in RC included *PDE4D*, whereas *SSBP2*, *KLF4*, and *GABRG2* were identified as potential targets in R-CRC.

*PDE4D* belongs to the subfamily four of cyclic nucleotide phosphodiesterase and has been implicated in the tumorigenesis of various neoplasms. Decreased expression of this gene has been found up to 30 times lower in samples from patients with chronic lymphocytic leukemia than in samples from healthy adults [29]. *PDE4D* also plays a significant role in CRC, where reduced expression has been associated with increased proliferation, colony formation, apoptosis, invasion, and migration [30]. It has been demonstrated that deregulation of *PDEA4* mediated by miR-203a-3p promotes tumor proliferation, growth, and cell migration in CRC [30]. Thus, the newly identified miRNA nov-miR-13844-5p may play a role in CRC by regulating the *PDE4D* gene, particularly in rectal tumors. Further functional studies are required to confirm these findings.

SSBP2 acts as a tumor suppressor by responding to DNA damage and promoting cell growth arrest by blocking the Wnt signaling pathway [31]. Reduced expression of this gene has been associated with worse prognosis in CRC and various other tumors, such as the esophagus, prostate, gallbladder, and acute leukemia [32]. The candidate nov-miR-13844-5p may play a regulatory role in the expression of this gene, particularly in R-CRC.

*KLF4*, a zinc finger transcription factor, plays a role in maintaining homeostasis in the intestinal epithelium [33,34]. *KLF4* is a tumor suppressor gene associated with cell proliferation, migration, and invasion. The mechanisms underlying *KLF4* deregulation include loss of heterozygosity, hypermethylation, and regulation by miRNAs. Various miRNAs have been implicated in KLF4 silencing in CRC cells. Taken together, these data indicate that nov-miR-13844-5p might be involved in colorectal carcinogenesis, making it a strong candidate for further validation.

The candidate, nov-miR-7154-5p, was significantly negatively correlated with *CADM2*, *EPHA5*, *CPED1*, *EPHA7*, *BEND6*, and *SERTAD4 expressions*. Among these, we highlighted that the tumor suppressor gene *CADM2*, which encodes a cell adhesion molecule, has been associated with prostate cancer, and its main mechanism of dysregulation is mediated by gene hypermethylation [35,36]. It has been reported that downregulation of *CADM2* mediated by miR-17-5p promotes a malignant phenotype in colorectal cancer [37]. In the current study, *CADM2* was found to be dysregulated in both R-CRC and R-CRC but not in RC.

The candidate nov-miR-590-5p was downregulated in CRC. Significantly lower levels of this miRNA were detected in L-CRC than in R-CRC tissues. Upregulation of *HOMER1* in L-CRC and *TBX1* in R-CRC, both predicted targets of this miRNA, was observed in these tumors. Overexpression of *HOMER1* has been reported in various tumors, including CRC [38]. The newly identified miR nov-miR-590-5p may be involved in *HOMER1* deregulation in CRC, particularly in right colon tumors, where a negative correlation was identified between this gene and the new miR. *TBX15*, identified as a target gene in L-CRC, is naturally more highly expressed during fetal development. In cancer, *TBX15* expression appears to be stimulated by NF-kB, exerting an anti-apoptotic effect and promoting cell proliferation [39]. Further studies are needed to ascertain whether *TBX15* acts as an oncogene in CRC, particularly in L-CRC, and whether it is regulated by the newly identified miR nov-miR-590.

Here, we showed an exploratory study to identify novel miRs using an in silico approach. Future directions include the detection of these novel miRs in colorectal samples and functional assays in CRC cell lineages to confirm their biological function in CRC.

In conclusion, this study marks a pivotal advancement in our understanding of colorectal cancer biology. The discovery of 15 novel microRNAs through rigorous analysis of small RNA sequencing data and the miRMaster algorithm underscores their potential significance. Validated against an independent dataset, these miRNAs have emerged as promising candidates for CRC treatment. Their ability to differentiate between tumor and normal tissues offers new prospects for diagnostics. Distinct expression patterns of miRNAs across different colon segments provide nuanced insights into CRC complexity. Additionally, the correlation of specific miRNAs with poor prognosis introduces potential prognostic biomarkers and refines patient care strategies.

Exploring miRNA-target gene interactions adds a layer of depth to elucidate potential mechanisms. The predicted associations with key genes involved in colorectal carcinogenesis suggest the impact of these miRNAs. This novel understanding sets the stage for innovative therapeutic approaches. The discovery of these miRNAs highlights the untapped potential of the human genome. As these findings intersect with established cancer biology, they open new avenues for diagnosis, prognosis, and treatment in colorectal cancer research and practice.

## 4. Materials and Methods

### 4.1. Study Cohorts

Discovery Group: Data from small-RNAseq sequencing of 522 colorectal samples (511 colorectal tumors and 11 adjacent normal tissues) were obtained from The Cancer Genome Atlas (TCGA) using the Cancer Genomics Hub (available online at: https://portal.gdc.cancer.gov/, accessed on 15 September 2020) GDC project ID: 24725. Clinical characteristics are summarized in Appendix A.

Validation Group: Publicly available small miRNA sequencing data from Gene Expression Omnibus (GEO) datasets (available online at: http://www.ncbi.nlm.nih.gov/gds, accessed on 30 September 2020) were used. The search was performed using the following keywords: colorectal cancer, non-coding RNA profiling by high-throughput sequencing, and Homo sapiens. The inclusion criteria were Illumina small RNA sequencing, a minimum of 25 CRC samples, and 5 MB of total reads. The study GSE89974, which followed all criteria, was included as the validation group and contained 35 CRC tissues and 20 normal samples.

### 4.2. Small-RNA Sequencing Processing and miRNA Discovery

A previously described discovery pipeline was used to identify novel unannotated miRNAs in CRC and normal samples [10,11,40]. Briefly, all raw sequences obtained from TCGA and GSE89974 datasets were processed. Raw BAM files were converted to the FASTQ format using the “SamToFastq” tool from the Picard analysis package (available online at: http://broadinstitute.github.io/picard/, accessed on 2 October 2020). Unaligned reads were trimmed based on their Phred quality score (≥20) and aligned to the latest version of the human genome (GRCh38) using spliced transcript alignment to a reference (STAR) aligner.

The previously unannotated miRNAs were identified using the miRMaster platform (available online at: https://ccb-compute.cs.uni-saarland.de/mirmaster/tutorial/, accessed on 2 October 2020), which, based on the FASTQ files, searches for new candidate miRNAs, quantifies their expression, and identifies miRNA isoforms and variants. The default parameters of miRMaster were used, followed by manual filtering. Sequences with low expression (RPM < 1 and expressed in less than 10% of the samples) were excluded. After filtering, the miRNA candidates were considered putative novel miRNAs. Both the discovery group (TCGA) and the validation group (GSE89974) were subjected to the same parameters and analysis (Figure 1).

For the identification of novel DE miRNAs in tumors, DEGs were considered upregulated and downregulated if FC = log2(Tumor/Normal +1.2) were ≥ 1.2 and FC = log2(Tumor/Normal +1.2) were ≤ 1.1, respectively. We considered validated putative novel miRNAs identified in both cohorts (discovery and validation groups) with the same expression pattern in CRC samples (upregulated or downregulated). Principal component analysis (PCA) was performed using the ClustVis platform [41] with both known and newly identified miRNAs.

### 4.3. Novel microRNA Analysis

The validated putative miRNA candidates were further subjected to five different analyses: structural analysis, tissue specificity, differential expression according to tumor location, survival analysis, and target prediction (Figure 1).

Structural analysis: The GC content and sequence length of putative novel miRNAs were evaluated using the Graph Pad Prism 8 program, and the fragile sites database HumCFS [42] was used to evaluate the genomic location of the miRNA candidates.

Tissue specificity: The newly identified miRNAs in CRC tissues were compared with those detected in normal tissues from different locations. This analysis included normal samples from TCGA and GSE89974 and adjacent normal tissues from five different anatomical sites obtained from TCGA: the lung (n = 89), breast (n = 90), urinary bladder (n = 19), liver (n = 59), and central nervous system (n = 5). Novel miRNAs were detected in these samples following the same pipeline described above from the raw small RNA sequences retrieved from TCGA. A t-distributed Stochastic Neighbor Embedding (t-SNE) nonlinear dimensionality reduction analysis was performed using the Orange Data Mining program [43]. A heatmap was constructed using the R ComplexHeatmap package using Z-score [44] and finalized in Inkscape (available online at: https://inkscape.org/, accessed on 10 October 2020).

Expression of novel miRNAs according to tumor location: The expression of the validated candidates was analyzed based on tumor location: right colon (cecum, ascending colon, hepatic flexure, and transverse colon), left colon (splenic flexure, descending colon, sigmoid junction, and rectosigmoid junction), and rectum. The Mann–Whitney and Kruskal–Wallis tests were applied.

Survival analysis: Cancer-specific survival curves for each of the validated candidate miRNAs were constructed based on the expression pattern of these miRNAs in tumor samples from TCGA. Expression data were dichotomized into normal and upregulated or downregulated expression according to z-score >2.0 or <−2.0, respectively, relative to normal tissues. Survival analysis was performed using the Kaplan–Meier method and log-rank test for curve comparison. Analyses were performed using the SPSS v27 program.

Prediction of targets regulated by candidate miRNA target genes was performed using the MirTarget tool [45] from the miRDB platform [46] using a score cutoff of ≥80. The expression levels of the target genes were further evaluated in the same tumor samples using RNA-seq data from the Xena Browser (available online at: https://xenabrowser.net/, accessed on 10 October 2020) [47]. Target genes with differential expression in TCGA-COAD (n = 567) and TCGA-READ (n = 97) were identified by comparing tumor and normal samples, considering *p* < 0.05 and FC ≥ 2 for upregulated and FC ≤ −2 for downregulated targets. Subsequently, paired miRNA targets were detected using opposite FC directions. In addition, Spearman correlation analysis was used to refine the most probable miRNA-target pairs (*p* < 0.05, r ≤ −0.1). Target prediction according to tumor location (right colon, left colon, and rectum) was performed following the same analyses.

### 4.4. Gene Network

The STRING platform (available online at: https://string-db.org/, accessed on 10 October 2020) [48] was used to identify interactions between target genes. The Cytoscape program [49] was used to construct interaction network figures showing target genes and their novel regulatory miRNAs.

## Figures and Tables

**Figure 1 ncrna-09-00065-f001:**
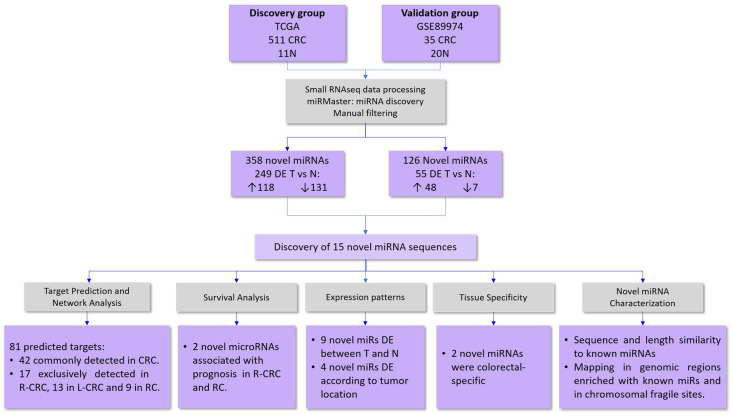
Flowchart describing the study design and main findings. Raw small RNA sequencing data from the discovery group (511 CRC and 11 N tissues) from the TCGA database and the validation group from the GEO database (35 CRC and 20 N, GSE89974) were processed and analyzed using the miRMaster tool. Curate analysis revealed 358 and 126 miRNA candidates in the discovery and validation groups, respectively. By comparing tumor versus normal tissues, 249 and 55 DE miRNA candidates were detected in CRC samples from the discovery and validation groups, respectively. Fifteen novel DE miRNAs were detected in both groups with the same expression patterns (upregulated or downregulated in tumor samples). The biological significance of these novel miRNA candidates was supported by similarities in structural and genomic mapping with known miRNAs, colorectal tissue specificity, association with tumor location and prognosis, and correlation with predicted cancer-related target genes. CRC, colorectal cancer; T, tumor; N, adjacent normal tissues; DE, differentially expressed; R-CRC, right-sided colorectal cancer; L-CRC, left-sided colorectal cancer; RC, rectal cancer.

**Figure 2 ncrna-09-00065-f002:**
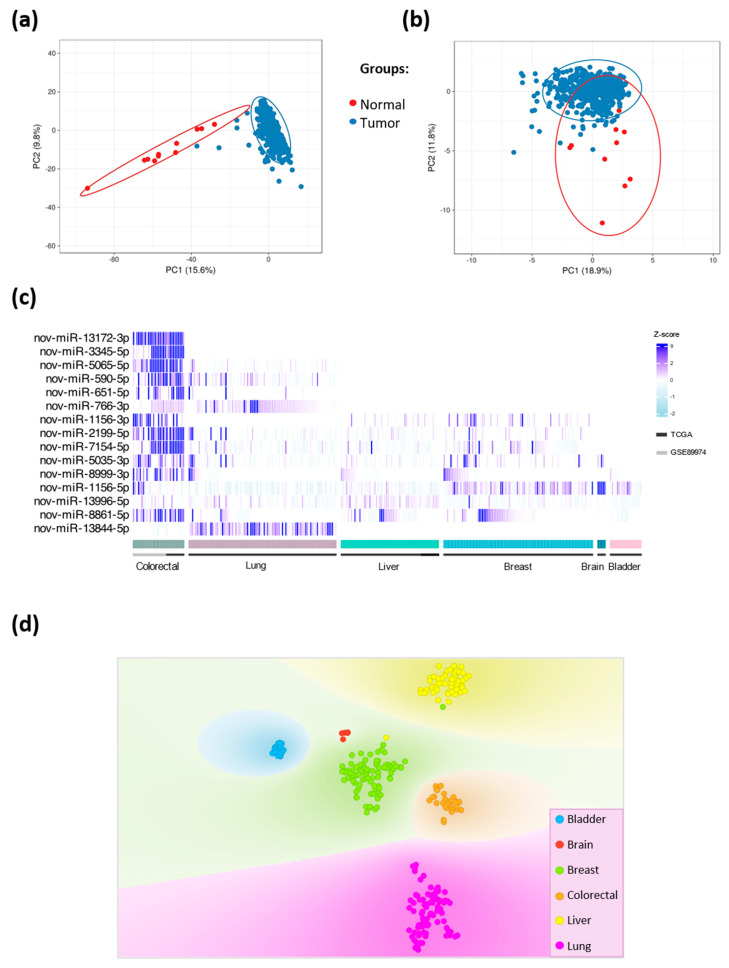
Tissue-specific expression patterns of novel miRNAs: (**a**) PCA of known miRNAs showing a clear separation between tumor and normal samples. (**b**) PCA showing tumor and normal samples separated according to the expression of novel miRNAs. (**c**) Heatmap showing the expression patterns of the 15 novel miRNAs in colorectal samples (left) and the five other non-malignant tissue types from TCGA. Two novel miRNAs (nov-miR-13172-3p and nov-miR-3345-5p) were exclusively found in colorectal samples, whereas three novel miRNAs (nov-miR-1156-5p, nov-miR-13996-5p, and nov-miR-8861-5p) were ubiquitously expressed in all tested samples. (**d**) t-distributed Stochastic Neighbor Embedding (t-SNE) analysis of novel miRNAs in normal colorectal samples (n = 31) from TCGA and GSE89974 and normal tissues from five different anatomical sites obtained from TCGA: lung (n = 89), breast (n = 90), urinary bladder (n = 19), liver (n = 59), and central nervous system (n = 5). The distribution of the samples in the chart demonstrates a cluster of colorectal samples (orange), clearly separated from the other non-malignant tissue types from TCGA, according to the 15 novel miR expression patterns.

**Figure 3 ncrna-09-00065-f003:**
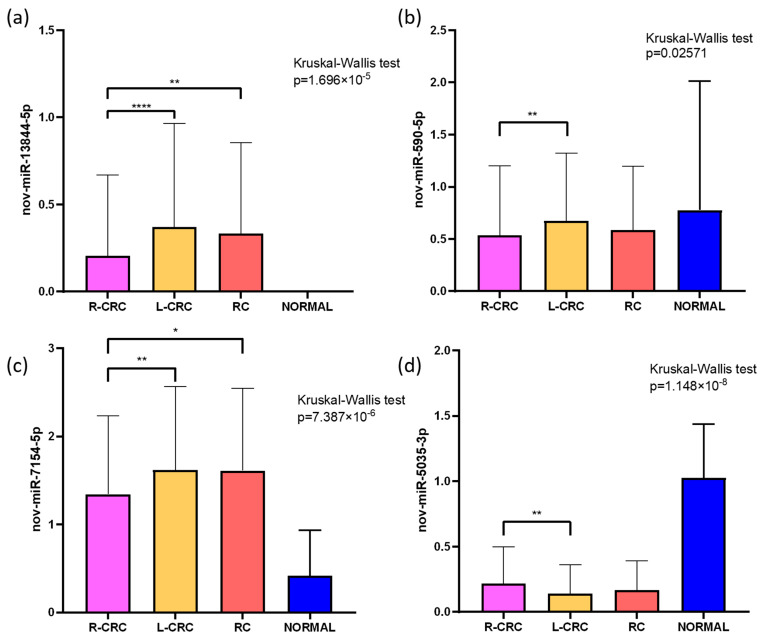
Expression patterns of the four novel microRNAs differentially expressed (DE) between colorectal anatomical sites of TCGA samples (* = *p* ≤ 0.05, ** = *p* ≤ 0.01, **** = *p* ≤ 0.0001): (**a**) Up-regulation of nov-miR-13844-5p was detected in tumors compared to normal samples, with higher levels detected in L-CRC and RC samples. This miRNA was not detected in normal samples from the TCGA cohort. (**b**) nov-miR-590-5p was more highly expressed in L-CRC. (**c**) Overexpression of nov-miR-7154-5p was observed in tumors compared with normal samples, with higher levels detected in L-CRC and RC compared to R-CRC tissues. (**d**) Downregulation of nov-miR-5035-3p was observed in tumors compared to normal samples, with lower levels detected in L-CRC compared to R-CRC samples. R-CRC, right-sided colorectal cancer; L-CRC, left-sided colorectal cancer; RC, rectal cancer. X-axis represents the log_2_FC of the novel microRNAs.

**Figure 4 ncrna-09-00065-f004:**
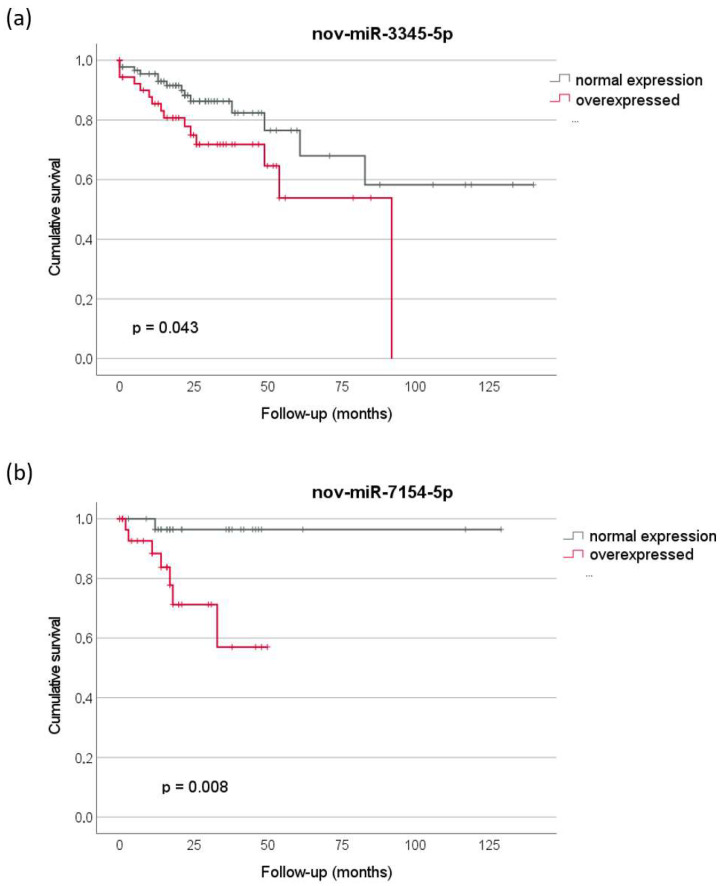
Cancer-specific survival curves according to the expression levels of (**a**) nov-miR-3345-5p in R-CRC and (**b**) nov-miR-7154-5p in RC, showing significantly decreased survival associated with overexpression of these novel miRNAs in these tumors. Log rank test. R-CRC: Right-sided colorectal cancer; RC: rectal cancer.

**Figure 5 ncrna-09-00065-f005:**
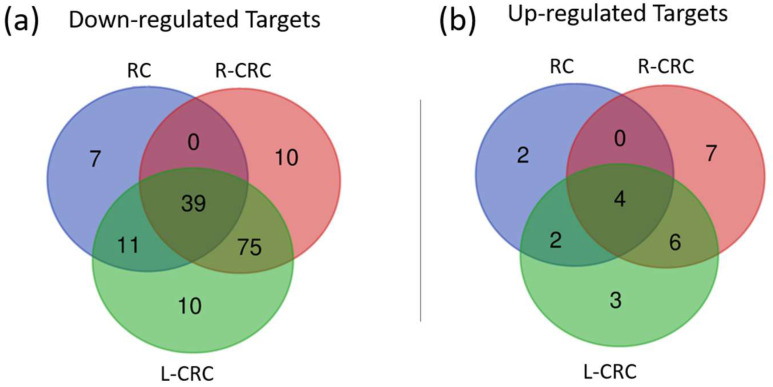
Venn Diagram showing (**a**) down-regulated and (**b**) upregulated predicted targets for right-sided colorectal cancer (R-CRC), left-sided colorectal cancer (L-CRC), and rectal cancer (RC). Upregulated and downregulated targets: FC ≥ 2, FC ≤ −2, respectively, and *p* < 0.05.

**Figure 6 ncrna-09-00065-f006:**
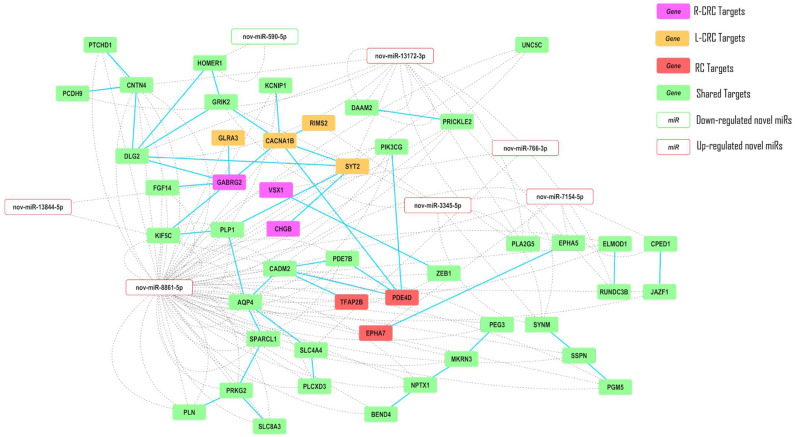
Interaction network among the genes predicted to be regulated by the candidate novel miRNAs with significant negative correlation and their regulators in the analyses of right-sided colorectal cancer (R-CRC), left-sided colon carcinomas (L-CRC), and rectal cancer (RC). Full blue line: gene–gene interaction; dotted gray line: novel miRNA–gene relationship.

**Table 1 ncrna-09-00065-t001:** Expression patterns of 21 candidates for new miRNAs identified in both discovery (TCGA) and validation (GSE89974) groups. In bold, 15 candidates with concordant fold changes (FC) in both groups.

Novel miRNA	Discovery Group FC ^1^	*p*-Value	Validation Group FC ^1^	*p*-Value
**nov-miR-13844-5p**	**- ^2^**	**4.15 × 10^−31^**	**2.4**	**1.83 × 10^−1^**
**nov-miR-766-3p**	**25.753**	**2.33 × 10^−7^**	**1.5**	**9.21 × 10^−2^**
**nov-miR-7154-5p**	**6.561**	**2.49 × 10^−5^**	**3.8**	**1.12 × 10^−6^**
**nov-miR-13172-3p**	**5.651**	**4.32 × 10^−4^**	**1.9**	**2.12 × 10^−3^**
**nov-miR-3345-5p**	**4.558**	**4.21 × 10^−5^**	**1.5**	**2.72 × 10^−2^**
**nov-miR-2199-5p**	**2.002**	**5.69 × 10^−1^**	**3.2**	**1.18 × 10^−4^**
**nov-miR-8861-5p**	**1.732**	**1.41 × 10^−1^**	**2.6**	**5.26 × 10^−3^**
**nov-miR-5035-3p**	**0.131**	**4.19 × 10^−5^**	**0.6**	**2.74 × 10^−1^**
**nov-miR-5065-5p**	**0.121**	**4.43 × 10^−4^**	**0.7**	**2.23 × 10^−1^**
**nov-miR-1156-3p**	**0.201**	**3.05 × 10^−3^**	**1.0**	**9.41 × 10^−1^**
**nov-miR-8999-3p**	**0.032**	**1.94 × 10^−2^**	**0.4**	**8.76 × 10^−2^**
**nov-miR-1156-5p**	**0.674**	**1.49 × 10^−1^**	**1.0**	**9.41 × 10^−1^**
**nov-miR-651-5p**	**0.671**	**7.35 × 10^−1^**	**0.9**	**2.70 × 10^−1^**
**nov-miR-590-5p**	**0.476**	**7.22 × 10^−1^**	**0.6**	**5.45 × 10^−2^**
**nov-miR-13996-5p**	**0.871**	**8.32 × 10^−1^**	**0.6**	**5.07 × 10^−1^**
nov-miR-9377-5p	15.757	3.46 × 10^−8^	1.1	5.89 × 10^−1^
nov-miR-6348-3p	0.789	2.18 × 10^−1^	5.0	2.10 × 10^−4^
nov-miR-13634-5p	2.784	1.77 × 10^−3^	0.8	4.09 × 10^−1^
nov-miR-171-3p	0.450	4.80 × 10^−3^	3.0	8.55 × 10^−2^
nov-miR-4285-5p	0.172	3.42 × 10^−8^	1.4	1.24 × 10^−1^
nov-miR-2549-5p	0.054	1.34 × 10^−4^	1.3	7.89 × 10^−1^

^1^ FC = log2(Tumor/Normal +1.2); ^2^ the candidate nov-miR-13844-5p was not detected in adjacent normal samples from the discovery group, although it was detected in normal samples from the discovery group.

## Data Availability

TCGA dataset: https://portal.gdc.cancer.gov/repository (accessed on 10 October 2020). GSE89974: https://www.ncbi.nlm.nih.gov/geo/query/acc.cgi?acc=GSE89974 (accessed on 10 October 2020).

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
