# Peer review of "Discovery of Novel miRNAs in Colorectal Cancer: Potential Biological Roles and Clinical Utility"

_ncrna, 2023, doi:10.3390/ncrna9060065_

Round 1
Reviewer 1 Report
Minutentag et al., aim to identify and explore the potential biological roles and clinical utility of novel miRNAs in colorectal cancer. To this end, they use the miRMaster program to examine TCGA and GEO data sets both in the discovery and validation groups. These analysis yield 15 novel miRNA candidates. They then exploit various bioinformatic tools to interrogate the potential correlation between the expression pattern of these novel miRNAs and the location, type or prognosis of cancer.
Overall all the work is well-designed and clearly presented. The discovery of novel miRNAs and their potential function and clinical use would be of interest to the researchers working in the field. I have the following suggestions before the work can be considered for publication.
Major points:
1. A cut-off value of 1.1 or 1.2 is too low in the identification of DE miRNAs. What to say, for example, about the functional significance of a miRNA that is differentially expressed 1.1-fold in a tumor sample?
2. I wonder if it would be possible to over-express or knockdown any of the novel miRNAs in a cell line and check its influence on the cell face or the expression of its target mRNA?
3. Please indicate in the figure captions the filtering criteria (e.g., cut-off values for DE).
4. Table 1. Please explain what is meant by (log(X+1.2))—what is X here?
Minor points:
1. Line 25, Please remove one of the periods.
2. Line 47, “22nt” should be “22 nt”.
3. Lines 49 and 52, please place a period after each sentence.
4. Figure 1, in the gray box, “Smal” should be “small”; also, “processoing” should be “processing”.
5. Line 101, 301 and 302 “CG” should be “GC”
6. Please avoid paragraphs composed of a single sentence. Perhaps just combine it with the previous or the following paragraph (e.g. lines 394-395). Please check the whole manuscript for such cases.
Minor typos.
Author Response
We would like to thank the reviewers for careful reading of the manuscript. The reviewers' thoughtful comments helped us refine the manuscript. The changes are highlighted in grey in the attached revised manuscript. A graphical abstract was included as suggested by the Managing Editor.
Reviewer 1:
Major Points:
- A cut-off value of 1.1 or 1.2 is too low in the identification of DE miRNAs. What to say, for example, about the functional significance of a miRNA that is differentially expressed 1.1-fold in a tumor sample?
The FC values presented on Table 1 are logged Fold Changes, actually. Therefore, a cut-off of log2FC=1.1 means more than two-fold of difference between tumors and normal tissues. We clarified this information in Table 1 legend and in the Material and Methos Section (lines 477-479).
In addition, we would like to clarify that as the main objective of this study was to identify novel miRNAs, we chose for a less stringent cut-off in the first filter to avoid losing potential new molecules. The lower cut-off values allowed us to identify a higher number of molecules for further analysis. We refined our selection in the subsequent analysis, specially by validating the identified novel miRNAs in an independent cohort of CRC patients presenting the same patterns of expression between tumors and normal tissues.
- I wonder if it would be possible to over-express or knockdown any of the novel miRNAs in a cell line and check its influence on the cell face or the expression of its target mRNA?
That is a very nice suggestion. We are aware that our work is an exploratory study, and therefore experimental studies are mandatory to prove the expression and biological relevance of these novel miRs in CRC. We are planning the next experiments in colorectal tissues (tumor and normal tissues) as well as functional validation in colorectal cell lines. We added a paragraph explaining these future directions in Discussion Section (lines 427-429).
- Please indicate in the figure captions the filtering criteria (e.g., cut-off values for DE).
We followed the suggestion and included the cut-off values in the legend of Figure 5.
- Table 1. Please explain what is meant by (log(X+1.2))—what is X here?
The X means tumor/normal ratio. We modified the Table Caption and included a legend. We also make modifications in the text of Material and Methods Section (lines 477-479).
Minor Points:
- Line 25, Please remove one of the periods.
We checked all manuscript for errors in punctuation. We removed the period as solicited.
- Line 47, “22nt” should be “22 nt”.
The error was adjusted as solicited.
- Lines 49 and 52, please place a period after each sentence.
We have placed the periods in lines 49 and 52 as solicited, in revised version lines 53 and 56.
- Figure 1, in the gray box, “Smal” should be “small”; also, “processoing” should be “processing”.
Both errors in the text of the grey boxes in Figure 1 were corrected.
- Line 101, 301 and 302 “CG” should be “GC”
We change all CG to GC across the paper accordingly.
- Please avoid paragraphs composed of a single sentence. Perhaps just combine it with the previous or the following paragraph (e.g. lines 394-395). Please check the whole manuscript for such cases.
We checked the whole manuscript and modified accordingly.

Reviewer 2 Report
Iael W. Minutentag et. al., have employed NGS approaches and have uncovered novel miRNAs in CRC. this work is relevant and interesting. The authors may choose to provide a future work direction that can stem using this discovery as a foundation.
Author Response
We would like to thank the reviewers for careful reading of the manuscript. The reviewers' thoughtful comments helped us refine the manuscript. The changes are highlighted in grey in the attached revised manuscript. A graphical abstract was included as suggested by the Managing Editor.
Reviewer 2:
Iael W. Minutentag et. al., have employed NGS approaches and have uncovered novel miRNAs in CRC. this work is relevant and interesting. The authors may choose to provide a future work direction that can stem using this discovery as a foundation.
Thank you for your suggestion. We added a paragraph about the study limitation and including the future work direction. It can be found in Discussion Section (lines 422 – 424).
